# Isolation and Identification Rust Pathogens and the Study of Antioxidant Enzyme Activity and Gene Expression under Rust Infection in *Zoysia japonica*

**Di Zhang** [1], **Hanguo Zeng** [1], **Liyuan Zhao** [1], **Jiaming Yue** [1], **Xiao Qi** [2] and **Manli Li** [1,*]

[1] College of Grassland Science and Technology, China Agricultural University, Beijing 100193, China; dichuengcau@163.com (D.Z.); zhgcau@163.com (H.Z.); 13717830620@163.com (L.Z.); ymm8093@163.com (J.Y.)

[2] National Animal Husbandary Service, Ministry of Agriculture and Rural Affairs, Beijing 100025, China; tq07mms@sina.com

\* Correspondence: lmlcau@126.com; Tel.: +86-010-62731264

**Abstract:** The goal of this study was to identify the zoysiagrass rust pathogens and to analyze the differences in rust-resistant and rust-susceptible *Zoysia japonica* germplasm upon inoculation. Based on the assessment of spore morphology and 18S ribosomal DNA (rDNA) molecular identification, the zoysiagrass rust pathogen was identified as *Puccinia zoysiae* Diet. The development of mycelium, the rate of spreading, and the timing of spore production were more delayed in the rust-resistant (RR) genotype than the rust-susceptible (RS) genotype. After inoculation, the activities of superoxide dismutase (SOD), catalase (CAT), and ascorbate peroxidase (APX) initially decreased, then increased in both the RR and RS genotypes, but the increased enzyme activities were faster in the RR than in the RS genotype. Rust resistance was positively correlated with antioxidant enzyme activity. The observed changes in CAT, POD and APX activity corresponded to their gene expression levels. The results of this study may be utilized in accurately evaluating the damage of rust disease and rust-resistance in zoysiagrass germplasm aimed at breeding the rust-resistant zoysiagrass varieties and improving disease management of zoysiagrass turf.

**Keywords:** zoysiagrass rust; 18S rDNA; antioxidant enzyme; gene expression; scanning electron microscope



## 1. Introduction

Rust is a common airborne fungal disease that can seriously reduce the turf quality, potentially causing huge economic losses [1]. The occurrence of rust disease is closely associated with the temperature and humidity. During an early disease onset, faded green spots occur at the infection sites where uredinia are formed and a large number of urediniospores are released. After plants are infected, infected cells or organs form galls or undergo necrosis. The rust pathogen absorbs nutrients from living plant cells, destroys chlorophyll, and reduces photosynthetic and respiratory rates [2], thus negatively affecting growth, physiology and overall health of the plants.

Proper identification of the genus and species of a rust fungus is a prerequisite to control rust disease. Because different fungi have variable sensitivity to fungicides, it is important to identify the fungal strain prior to effectively control the disease. Currently, studies on fungal strains mostly include morphological and molecular identifications. The morphological identification of rust pathogens is generally based on the shape, size, and color of the spores [3]. However, due to morphological similarity of many fungi species, distinguishing the taxonomic status of some rust fungi may be challenging when only based on fungi morphological features. Alternatively, molecular identification of fungi can be an effective way in differentiating fungal species. Molecular markers are simple and highly accurate methods that have been utilized in classification and phylogeny studies [3].

Specifically, molecular identification using 18SrDNA has advantages due to its rapid, accurate, trace and convenient assessment. This method offers abundant genetic information for the classification and identification of fungi in the systematics research of fungi genus and species [4]. A previous study showed that combining morphological observation and 18S rDNA sequence information is the most helpful approach to identification of fungal species [5].

Several plant anatomical features can affect rust infection including the thickness of the cuticle, phellem, and cell wall, the number, size, and stomatal aperture, and the existence of the wax layer on leaf surface [6,7]. For example, Li [8] observed that the extent of the wax layer in the rust-susceptible *Zoysia japonica* was about 30% lower than the rust-resistant one. The number of stomata in the rust-resistant genotype was 68% higher than that in the susceptible genotype. Moreover, the stomata were shorter and narrower and the thickness of the spongy mesophyll cell layer and the palisade mesophyll cell lyer were higher in the rust-resistant genotype.

Plants and rust fungi have always coexisted in nature. Through the process of selection and adaptation, host plants gradually improve their disease resistance mechanisms. Superoxide dismutase (SOD), catalase (CAT), ascorbate peroxidase (APX) and peroxidase (POD) are antioxidant enzymes that can be used as biochemical markers of cellular defense system against oxidative damage, contributing to plant stress resistance [9]. Polyphenol oxidase (PPO) plays an important role in plant resistance to external stress, photosynthesis and biosynthesis [10]. After rust fungi inoculation of adzuki bean varieties carrying different levels of rust resistance, activities of SOD, CAT, POD, and PPO changed more significantly in the resistant cultivars [11]. At 48 h after inoculation, the activity of SOD and PPO was higher in the resistant variety than that in the susceptible variety. These results suggest that antioxidant enzymes play a role in conferring rust resistance. However, it is unclear whether alterations of antioxidant enzymes are associated with rust resistance in *Z. japonica*.

*Z. japonica* is a perennial herbaceous plant found in temperate regions with a dense root system, dense and hard leaves, and extensive creeping stolons [12]. *Z. japonica* exhibits a strong propensity to develop tillers and is highly adaptive to drought and environments with poor soil fertility. Because of its tolerance to short mowing, *Z. japonica* is widely used on sports fields and urban green lawns [12]. This species is mostly distributed in the coastal areas, including New Zealand to Hokkaido, Japan, Southeast Asian countries, and Australia [13]. In China, it is found in the coastal areas from the Liaodong Peninsula in the north to Fujian and Hainan in the south [14]. Rust is the most common, long-lasting disease in *Z. japonica*. Up until now, most studies have shown that rust disease in *Z. japonica* is caused by *Puccinia zoysiae* Diet [2], which has also been reported in *Zoysia tenuifolia* and *Z. japonica* [2,15,16].

Until now, studies on the identification of zoysiagrass rust pathogen have been limited to morphological observations. Various levels of resistance upon rust pathogen infection are not fully understood in zoysiagrass germplasm, especially those physiological and molecular mechanisms associated with resistance. In order to clarify the differences of physiological and biochemical in different degrees of resistance materials after rust fungus inoculated, to understand the response of zoysiagrass germplasm to rust inoculation, and to provide ideas for the control of rust, we conducted the following study. We observed the morphological characteristics of rust fungi combined with 18S rDNA sequence analysis to identify the zoysiagrass rust pathogen. We inoculated rust-susceptible and rust-resistant materials with the rust fungus, investigated the infection type, disease incidence rate, and disease index; and furthermore, measured the antioxidant enzyme activity and gene expression during the corresponding stages of rust disease. The results will ultimately help to reveal rust resistant mechanisms and benefit turf grass management controlling zoysiagrass rust and guide breeding programs for creating rust-resistant varieties.

## 2. Materials and Methods

### 2.1. Isolation and Identification of Zoysiagrass Rust Pathogen

Plant Materials and Inoculation. Seedlings of common Qingdao *Z. japonica* (10 cm in height) were used for inoculation with rust pathogen. Infected leaves were collected from Jiaozhou, Shandong and subjected to shade-drying for 3 to 4 days. Spores were collected and used to prepare a spore suspension as pathogen donor. The seeds used in the experiments were from the Qingdao Haiyuan Co., Ltd. (Qingdao, China) in 2017. The seeds were planted in an artificial climate chamber (25 °C–30 °C, 70% humidity, 600 μmol m$^{-2}$s$^{-1}$, 8 h light and 16 h dark) in September 2017. The seedling facility consisted of a substrate ratio of Pindstrup substrate and vermiculite equal to 2:1 in a plastic flowerpot of 18 cm height and 25 cm width; 100 seeds were randomly planted in each pot with four replicates, and watered daily. After 60 days growth, to about 10 cm in height, spray inoculation of the plants was performed. The concentration of spore suspension was adjusted to $5 \times 10^5$ mL$^{-1}$ with a hemocytometer [17]. The spore suspension (containing 0.2% gelatin) was sprayed on the seedlings for inoculation. Pots were covered with plastic bags, filled with air, closed with rubber bands, and then placed in an illuminating incubator for 24 h at 28 to 30 °C followed by 2 to 3 days in a moist chamber. Rust fungal powder was collected from infected seedlings and was stored at 4 °C for a short-term morphological observation and inoculation. The fungal powder was then stored at −20 °C for a longer period to allow molecular identification.

Morphological identification of rust pathogen. Rust fungus was cultured and observed for 15 days. On a clean bench, spores were picked with a needle, placed on a slide in a drop of sterile water, and observed using a light microscope (Olympus BX53). The color, shape, organization of the spores were recorded, and the size of the spores was measured using a micrometer. Microscopic images were taken. Morphological identification was conducted using the index of Pucciniales in *Flora Fungorum Sinicorum* (Volume 10) as a reference [18].

Molecular identification of rust pathogen. (i) PCR amplification. DNA was extracted from zoysiagrassrust pathogen using a DNA extraction kit (Aidlab Biotechnologies, Beijing, China), following the manufacturer's recommendations. Primers were synthesized by Tianyi Huiyuan Inc. (Beijing, China) and primer sequences are listed in Annexed Table 1 [19]. The primer sequences were conserved at the 5′ and 3′ end of the 18S rDNA. (ii) Sequencing and sequence alignment. PCR amplification products with three replicates were sequenced for single-end sequencing. Sequence alignment was performed using BLAST of NCBI (http://www.ncbi.nlm.nih.gov accession No: MH268241, accessed on 15 October 2021). Mega 5.0 software was used for multiple sequence alignment. A phylogenetic tree was constructed using the neighbor-joining method and tested using 1000 bootstrap replicates.

**Table 1.** Mean values for rust disease assessment in the resistant (RR) and susceptible (RS) genotype of *Zoysia japonica*.

| Material | Source | Incidence Rate (%) | Disease Indexes | Incubation Period (d) | Rust Severity | Infection Type of Material |
|----------|--------|-------------------|-----------------|----------------------|---------------|---------------------------|
| RS | Liaoning | 21.9 * | 3.90 * | 16–17 | 4 | 4 |
| RR | Shandong | 0 * | 0 * | - | 0 | 0 |

* Significant differences at $p < 0.05$.

### 2.2. Infection of Zoysiagrass Rust Pathogen in Different Germplasm of Z. japonica

The infected leaves of previously identified rust-resistant zoysiagrass were dried under shade for 3 to 4 days. The rust fungi were collected and used in preparing a spore suspension, which was utilized as pathogen donor. From 2008 to 2009, a total of 162 *Z. japonica* germplasm collected from various locations were initially tested under natural conditions or indoor inoculation. SD−1, SD−2, and SD−3 were preliminarily identified as the rust-resistant populations [16] (Zhang 2010). In 2011, a test involving multiple locations

was conducted on over the selected 40 *Z. japonica* materials obtained from the preliminary assessment [20]. In 2012, a subsequent screening experiment identified two germplasm materials with stable and consistent results: the rust-resistant material (RR) from Liaoning province and the rust-susceptible material (RS) from Shandong province. Four tillers of each genotype were selected from the same plant, transplanted into small pots (15 cm diameter, 20 cm height), and placed in the greenhouse for 45 to 60 days to obtain uniformly grown, healthy plants with good vigor. Both RR and RS materials had three replicates.

Assessment of disease resistance. Fungal powder was collected from the infected leaves of seedlings and used for inoculating RR and RS materials. The inoculation method is described above. A grading system was used to rank the disease occurrence during the peak of the rust disease. The assessment indices used for disease severity were the percentage of infected leaves, severity degree, incubation period, and infection type [21]. The following incubation measurements were recorded: (1) incubation period, time needed from inoculation to the first day of disease occurrence in one replicate and the disease occurrence in all replicates (Technical Specifications for Assessing and Forecast of Wheat Stripe Rust GB/T1595-1995); (2) incidence rate, the percentage of the number of infected leaves among all leaves; (3) severity degree, scores within the range of 0 to 7 were given based on the incidence rate and the severity of individual leaves; (4) disease index, sum of (number of plants at a severity degree × the severity degree) for all severity degrees (total number of plants × highest severity degree) [22].

We assessed infection type of zoysiagrass rust according to the Technical Specifications for Assessing and Forecast of Wheat Stripe Rust and Research Methods for Plant Diseases [22], with modifications based on the disease characteristics of *Z. japonica*. The severity of classification standard of zoysiagrass rust is shown in Supplementary Table S2 and Supplementary Figure S1. The infection type of classification standard of zoysiagrass rust is presented in Supplementary Table S3 and Supplementary Figure S2.

Microscopic observation. For scanning electron microscope (SEM), samples were collected at 0, 0.5, 1, 1.5, 3, 5, 10 15, and 20 days of post inoculation (dpi).

### 2.3. Antioxidant Enzyme Activity and Gene Expression Analyses

Ten leaves that grow uniformly from the RR and RS genotype which mentioned in 2.2 were collected at 0, 5, 10, 15, and 20 dpi with three random repetitions in three plastic flowerpots for initial sampling. The middle portions of the leaves with no uredinia were then cut into 0.2 cm pieces, mixed, and 0.1 g of leaf tissue was taken for extraction. The uninoculated, healthy leaves were used as the control. The activities of SOD, CAT, POD, and APX were measured three times for all samples. The activity of SOD was measured by the method described by Zhang and Kirkham [23]. The activities of CAT and POD were determined by the method of Chance and Maehly [24], and APX activity was measured according to the method of Nakano and Asada [25], with modifications.

Total RNA was extracted by using Tiangen DP432-RNA Prep Pure plant total RNA extraction kit, following the manufacturer's instructions. Standard agarose gel electrophoresis was run to detect RNA integrity (1.2% agarose, 0.5 × TBE buffer, 150 v, 15 min). The ratio of OD260/OD280 was used to evaluate RNA purity. RNA concentration was calculated as follows: final concentration (ng/μL) = (OD260) × (fold of dilution) × 40. Reverse transcription was performed using the PrimeScript$^{TM}$ RT Reagent Kit with gDNA Eraser (TaKaRa), following the manufacturer's instructions. The primer sequences used in this study are listed in Supplementary Table S4. For real-time quantitative PCR (qPCR), *Zjactin1* was used as internal reference for the analysis of differential expression. Rotor-Gene TM SYBR Green PCR Kit (TaKaRa) was used for qPCR. cDNA was diluted 10-fold for qRT-PCR analysis, The relative quantifification ($2^{-\Delta\Delta CT}$) of target gene expression was calculated using the comparative cycle threshold method [26].

### 2.4. Statistical Analysis

The significance of enzyme activity was analyzed with SPSS 18.0 (SPSS. Inc., Chicago, IL, USA). Paired *t*-test was conducted to analyze the data from different treatments with the same sampling time and *Z. japonica* material. Differences were deemed significant at $p < 0.05$.

## 3. Results

### 3.1. Morphological Identification of Zoysiagrass Rust Pathogen

As shown in Figure 1A,B, urediniospores are yellow or orange, obovate or spherical, with small spikes on the surface and a range of size from 18 to 22 × 14 to 17 μm. The spore wall had a uniform thickness of 2 to 2.4 μm with no color or light yellow. Under the microscope, the uredinia were orange-colored with paraphyses, extending parallel to the veins and growing in between the mesophyll cells. Uredinia were scattered and distributed as spots or bundles and bulging. When the spores were mature, the leaf surface broke open and yellow powder was released (Figure 2). Based on zoysiagrass rust symptoms and the morphological features of the rust pathogen under the microscope, we used the *Flora Fungorum Sinicorum* (Volume 10) as a reference and preliminarily identified the zoysiagrass rust pathogen as *P. zoysiae* Diet.

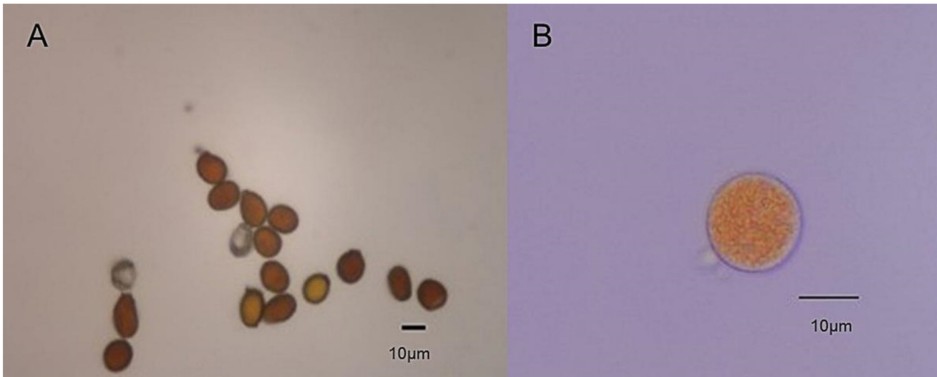

**Figure 1.** Spore morphology and color. (**A**) observed under the 10 × objective; (**B**) observed under the 40 × objective.

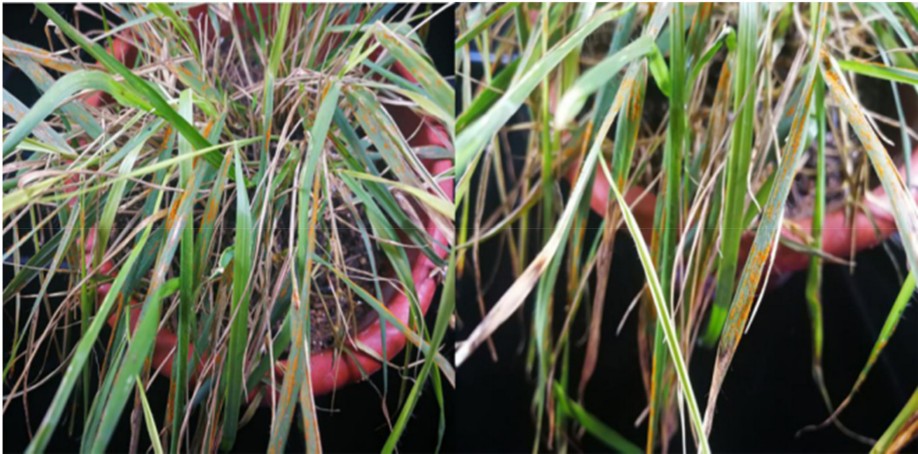

**Figure 2.** Symptoms of suscepible leaves in *Zoysia japonica*.

### 3.2. Sequence Homology Analysis of Zoysiagrass Rust Pathogen 18S rDNA

The rust pathogen collected from the infected leaves of Qingdao Z. japonica seedlings was recorded as 1811. As shown in Supplementary Figure S3A, a 1.3 kb specific fragment

was obtained. The cloning and assembly showed that the 18S rDNA of 1811 was 1309 bp in length. We aligned the sequence with the available Pucciniales sequences and found that the 18S rDNA sequence of the zoysiagrass rust pathogen had 96 to 99% homology with that of Uropyxidaceae and a 97 to 99% homology with that of Pucciniaceae. The zoysiagrass rust pathogen had a different morphology than Macruropyxis fraxini. However, 1811 was not closely related to the other families in Pucciniales and belonged to a different clade (Supplementary Figure S4). Sequence comparison revealed that the 18S rDNA sequence of zoysiagrass rust pathogen 1811 was closely related to that of Macruropyxis fraxini (KP858144.1) and P. physalidis (DQ354523.1) and was in the same clade as Macruropyxis fraxini (Supplementary Figures S5 and S6).

### 3.3. The Disease Severity and Infection Type Survey

The comparison of disease incidence in different Z. japonica materials is shown in Table 1. The incidence rate was lower in RR than in RS genotype. The length of the incubation period was shorter in RS than in RR. For degne, nearly immune, resistant, moderately resistant, moderately susceptible, and highly suscree of severity, RS was severe, whereas RR showed no symptoms. The disease index was lower in RR than in RS. The incidence was significantly differed in RR and RS ($p < 0.05$). After inoculation, RS genotype exhibited rust symptoms; spotted, yellow aeciospores appeared at 13 dpi. By 16 dpi, all replicates showed disease symptoms in RS. The RR genotype did not show any symptoms of infection or yellow spores after inoculation, and the plants grew well with many green leaves. Based on the disease resistance performance of the germplasm, the infection type can be classified as immueptible. The RR genotype was not infected after inoculation. Thus, the infection type of this material was immune, while RS genotype was highly susceptible after inoculation (Table 1).

### 3.4. Morphological Observations of Z. japonica Leaves Inoculated with Rust Pathogen

Before inoculation, the leaf surface, trichomes, and stomata exhibited plump shape in both genotypes (Figures 3 and 4). The leaf epidermal cells were well organized; stomata had a uniform size, closed, and bulged above the surface. At 1 dpi, the pathogen was not found on the leaves for both RR and RS genotypes, but some spores began to germinate. The germ tubes were 14.38 μm in length, 11.1 μm in width; germ tubes had a length of 1.68 μm in RR while a depression on the spores and germ pores had appeared and a few conidiospores were found on the spore surface in RS (Figures 3B and 4B). At 1.5 dpi, the mycelia had adhered to the leaf surface and surrounded by 30 to 35 conidiospores in RR, but in RS, mycelia had adhered onto the surface. The mycelia had elongated but did not increase in number (Figures 3C and 4C). At 3 dpi, the rust mycelia had spread throughout the entire host leaf surface as a web both in RR and RS, and conidiophores had formed. In RS, the mycelia were thicker and more extensive, with spores attached, while the conidiophores were not observed in RR (Figures 3D and 4D). In RR, at 5 dpi, approximately 15 to 20 conidiospores were noted (Figure 3E), while at 10 dpi, broken mycelia were found (Figure 3F), and conidiospores were still observed at 15 dpi (Figure 3G). In RS, at 5 dpi, numerous cylindrically shaped, vertically partitioned, unbranched conidiophores were found on the leaf surface with a few conidiospores scattered nearby (Figure 4E). For RS at 10 dpi, most spores in the uredinia had not completely developed (Figure 4F), while at 15 dpi, the spores in the uredinia were plump, densely distributed, with 26.1 μm in length and 26.5 μm in width (Figure 4G). For RR at 20 dpi, the germ tubes had formed by the spores with a length of 14.1 μm, a width of 11.65 μm; and the germ tube length of 12.25 μm. The surface of leaf tissues had minor damage, and ruptured cells were found around the spores in RR (Figure 3H). In RS, short germ tubes had formed from the spores scattered out of the uredinia, with the spore length of 13.4 μm, a width of 12 μm, and germ tubes length of 12.9 μm (Figure 4H).

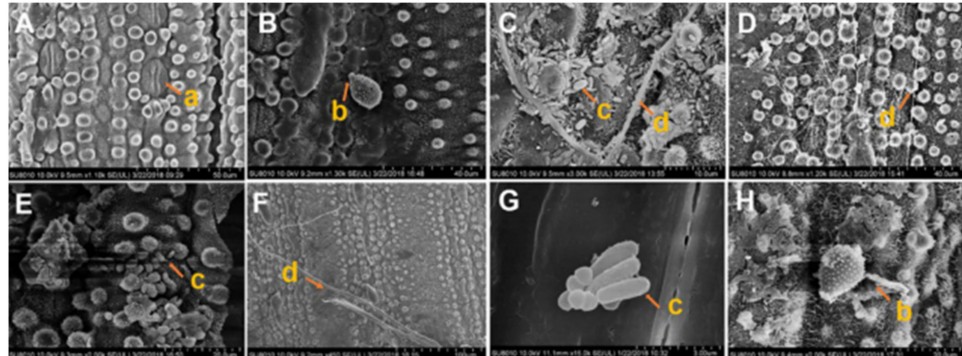

**Figure 3.** Observation of the leaves of the rust-resistant Zoysia japonica inoculated with puccinia by SEM. (**A**) no inoculation; (**B**) 1 day post inoculation (dpi); (**C**) 1.5 dpi; (**D**) 3 dpi; (**E**) 5 dpi; (**F**) 10 dpi; (**G**) 15 dpi; and (**H**) 20 dpi.

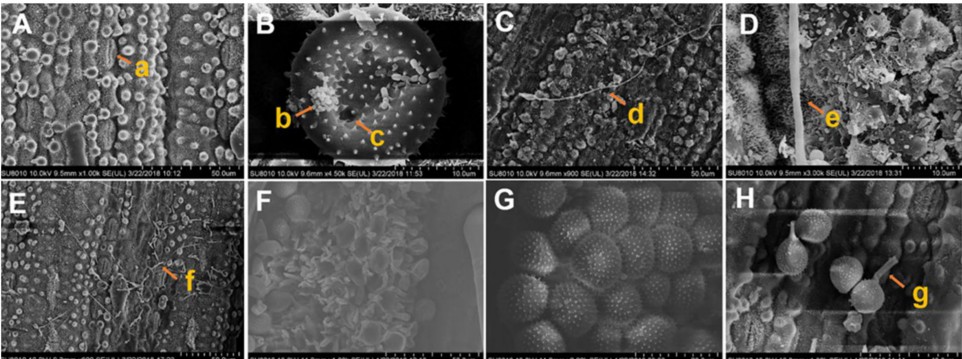

**Figure 4.** Observation of the leaves of the rust-susceptible Zoysia japonica inoculated with puccinia by SEM. (**A**) no inoculation; (**B**) 1 day post inoculation (dpi); (**C**) 1.5 dpi; (**D**) 3 dpi; (**E**) 5 dpi; (**F**) 10 dpi; (**G**) 15 dpi; and (**H**) 20 dpi.

The leaves were green with normal epidermis in RR from 0 to 15 dpi (Figure 5A–D). At 20 dpi, the green color of the leaf had slightly faded for RR (Figure 5E). For RS, the leaves were green and the epidermis was normal from 0 to 5 dpi (Figure 5A,B). However, the green color of the leaf had slightly faded in RS at 10 dpi (Figure 5C). From 15 to 20 dpi, uredinia were observed on the leaves of RS. The amount of uredinia and the area they covered gradually increased as the disease became more severe in RS (Figure 5D,E).

### 3.5. Antioxidant Enzyme Activity

Genotype had significant effects on SOD activity at 5, 15 and 20 dpi, while treatment effects on SOD activity were significant expect for at 0 dpi (Table 2). No genotype × treatment interactions were found for SOD activity until 20 dpi (Table 2). Rust infection differentially affected SOD activity in *Z. japonica* genotypes varying in rust resistance (Figure 6A). Activity of SOD initially decreased, then increased, reaching the lowest level at 10 dpi in both genotypes after inoculation (Figure 6A). With the extended period of inoculation, the SOD activity continuously increased, but such a trend was not observed in both non-inoculated genotypes. At 15 and 20 dpi, SOD activity in both inoculated genotypes was significantly higher than the non-inoculated plants ($p < 0.05$). Genotype had significant effects on CAT activity at 10 and 20 dpi, and treatment effects on CAT activity were significant expect for at 0 dpi (Table 2). There were no genotype × treatment interactions for CAT activity (Table 2). In inoculated plants, CAT activity initially decreased, then increased, reaching its lowest level at 5 dpi in both genotypes (Figure 6B). In non-inoculated plants, CAT activity remained stable during the early stages, whereas in the later stages, CAT activity decreased then increased, reaching its lowest level at 15 dpi

(Figure 6B). Genotype had significant effects on POD activity only at 5 dpi, and treatment effects on POD activity were significant at 5, 15 and 20 dpi (Table 2). Significant genotype × treatment interactions were shown at 5, 15 and 20 dpi (Table 2). In inoculated plants, the POD activity was at the lowest level at 5 dpi in both genotypes (Figure 6C). After that, POD activity rapidly increased and reached its peak. Genotype had significant effects on APX activity at 5, 10 and 15 dpi, and treatment effects were significant expect for 0 dpi (Table 2). Significant genotype × treatment interactions were found at 5, 10, and 15 dpi (Table 2). The APX activity first decreased, then increased, reaching its lowest level at 5 dpi in inoculated in both genotypes (Figure 6D). In uninoculated plants, APX activity decreased and reached its lowest level at 15 dpi.

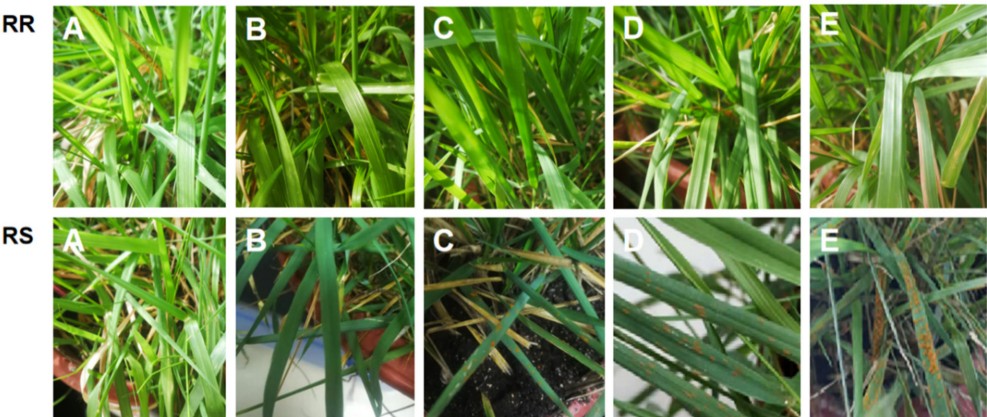

**Figure 5.** Observation of the leaves of the rust-resistant Zoysia japonica (RR) and rust-susceptible Zoysia japonica (RS) inoculated with puccinia. (**A**) 1 day post inoculation (dpi); (**B**) 5 dpi; (**C**) 10 dpi; (**D**) 15 dpi; and (**E**) 20 dpi.

**Table 2.** ANOVA results of superoxide dismutase (SOD), catalase (CAT), peroxidase (POD), and ascorbate peroxidase (APX) activity of *Zoysia japonica* after inoculation with the rust pathogen.

|  | 0 Days | 5 Days | 10 Days | 15 Days | 20 Days |
|---|---|---|---|---|---|
| SOD |  |  |  |  |  |
| Genotype (G) | ns | ** | ns | * | ** |
| Treatment (T) | ns | ** | ** | ** | ** |
| G × T | ns | ns | ns | ns | * |
| CAT |  |  |  |  |  |
| Genotype (G) | ns | ns | * | ns | ** |
| Treatment (T) | ns | ** | ** | ** | ** |
| G × T | ns | ns | ns | ns | ns |
| POD |  |  |  |  |  |
| Genotype (G) | ns | ** | ns | ns | ns |
| Treatment (T) | ns | ** | ns | * | ** |
| G × T | ns | ** | ns | ** | ** |
| APX |  |  |  |  |  |
| Genotype (G) | ns | * | ** | ** | ns |
| Treatment (T) | ns | ** | ** | ** | ** |
| G × T | ns | ** | ** | ** | ns |

* and ** represent significance at $p < 0.05$ and 0.01; ns, no significance.

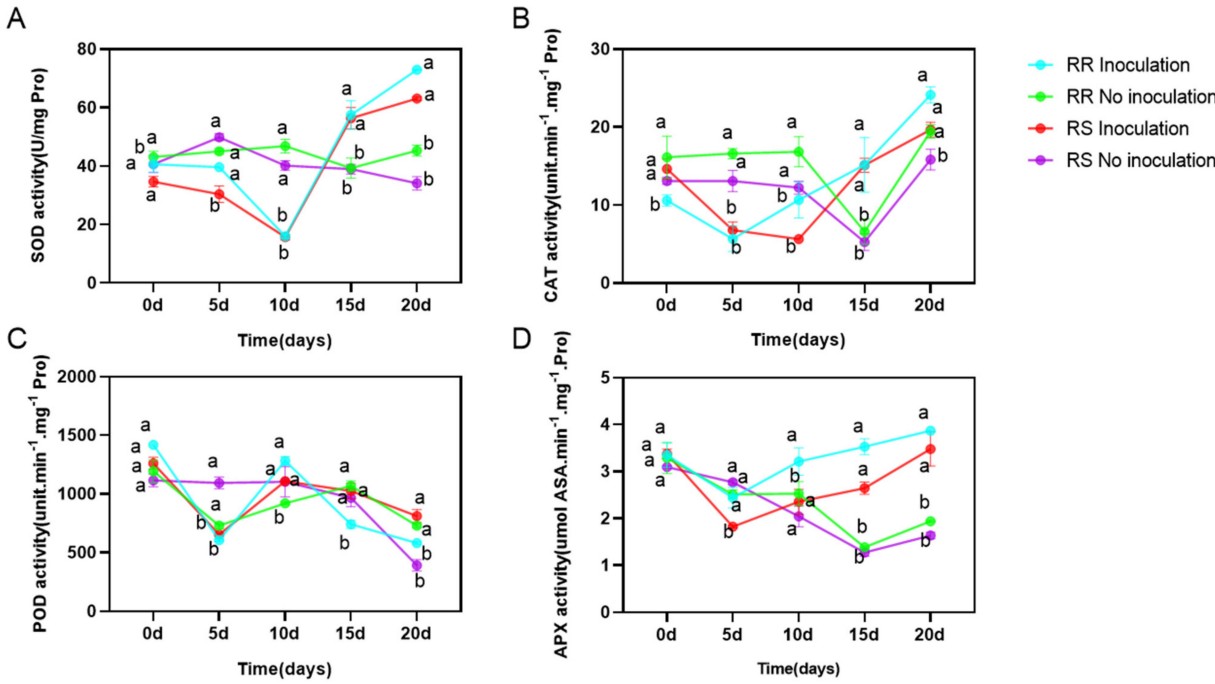

**Figure 6.** Effect of inoculation of puccinia on superoxide dismutase (SOD), catalase (CAT), peroxidase (POD) and ascorbate peroxidase (APX) activity of *Zoysia japonica*. Comparisons were made among rust resistant genotype (RR) and rust sensitive genotype (RS) under non-inoculation and after inoculation at a given day. Each value represents the mean ± standard error (SE) of three replicates. Means followed by same letter are not significantly different at $p < 0.05$.

### 3.6. Gene Expression Analysis

Based on the data of antioxidant enzyme activity in the two genotypes, we selected leaf samples at 5 dpi for gene expression analysis using RT-qPCR, and the results of differential gene expression analysis are shown in Figure 7. The inoculated plants had significantly higher *ChlCu/ZnSOD* but lower *MnSOD* expression levels than the non-inoculated plants. The expression levels of *ChlCu/ZnSOD* and *MnSOD* gene were upregulated in the RR and downregulated in the RS genotype. The expression of *POD* in the inoculated plants was significantly lower than that of non-inoculated plants. The gene expression levels of *CAT* and *POD* were downregulated in both genotypes after inoculation. Expression of the *APX* gene in the inoculated plants was significantly lower than that in non-inoculated plants. This gene expression was upregulated in the RR and downregulated in the RS after inoculation.

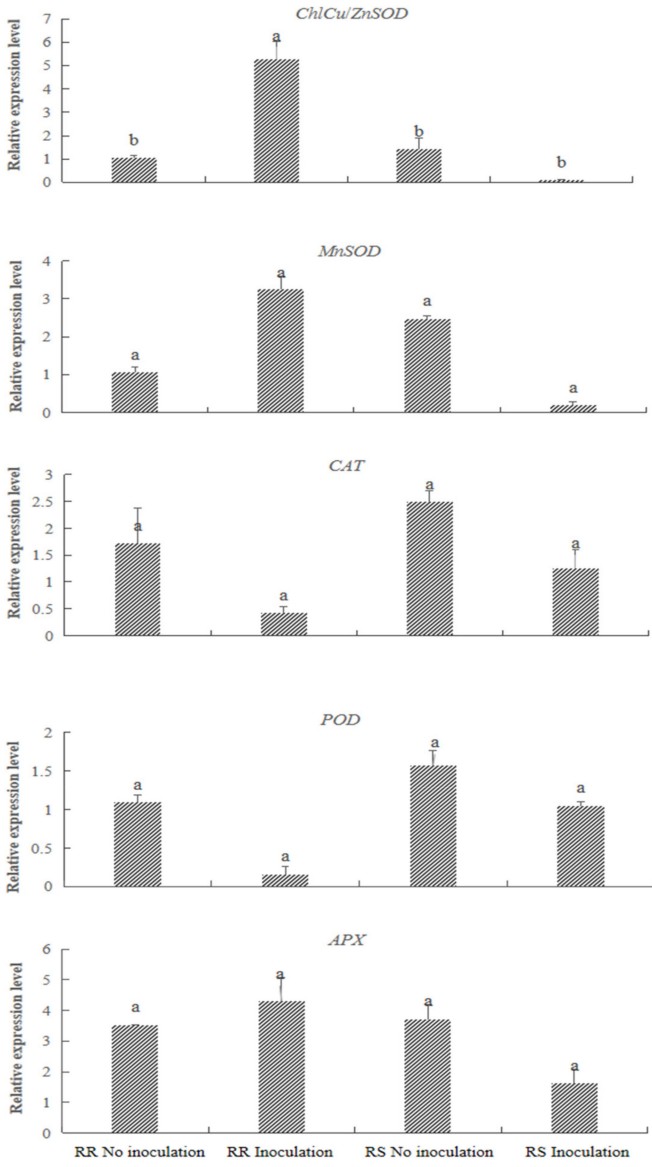

**Figure 7.** Relative expression levels of genes encoding chloroplastic superoxide dismutase (*Chl Cu/Zn SOD*), manganese SOD (*MnSOD*), catalase (*CAT*), peroxidase (*POD*), and ascorbate peroxidase (*APX*). Comparisons were made among rust resistant genotype (RR) and rust sensitive genotype (RS) after 5 days of non-inoculation and inoculation. Each value represents the mean ± standard error (SE) of three replicates. Means followed by same letter are not significantly different at *p* < 0.05.

## 4. Discussion

Identification of the zoysiagrass rust pathogen. Identification of the pathogenic fungal strain is beneficial for selecting effective fungicide for disease control and serves as the foundation for epidemiological investigations [27]. Previously, Li [28] isolated and purified the pathogen from rust-infected hyacinth leaves, performed 18S rDNA sequence homology analysis, and identified the rust pathogen as *Uromyces muscari*. In this study, we found that the urediniospores of zoysiagrass rust were orange in color, with paraphyses, spreading parallel to the veins, and growing between mesophyll cells. The uredinia were scattered and distributed as spots or bundles, bulging out of the surface; after maturation, leaf epidermis ruptured and yellow powder was dispersed outwards (Figures 1 and 2). This was consistent with the results from previous studies on rust pathogen identifications in zoysiagrass [2,29]. For pathogen identification, traditional methods are widely used based on the morphology of urediniospores and teliospores. However, the use of the

traditional classification method may cause uncertainty due to high similarity in spore morphology of different rust pathogens. Combining the traditional method with molecular identification can further improve the accuracy in rust identification [30]. The spore morphology of the rust fungus from this study was consistent with the *P. zoysiae* Diet spores described in the *Flora Fungorum Sinicorum* (Volume 10) [18]. In addition, 18S rDNA sequence revealed that zoysiagrass rust pathogen was 1309 bp in length and was distantly related to other families, while exhibiting the closest homology with *Macruropyxis fraxini*. The morphological features of *M. fraxini* were significantly different from those of the zoysiagrass rust pathogen in this study (Supplementary Figures S5 and S6). Therefore, the zoysiagrass rust pathogen in the Jiaozhou area was identified as *P. zoysiae* Diet, consistent with the results from previous studies on rust pathogen identifications [29,31].

Disease resistance assessment after inoculation. After inoculation, the RR genotype did not exhibit any disease symptoms during the observation period, indicating that it was resistant to zoysiagrass rust. In contrast, the RS genotype was highly susceptible to zoysiagrass rust. The incidence rate and disease index significantly differed between the RR and RS plants (Table 1 and Figure 5). Our results supported a previous field observation that the disease index of RR and RS plants was 0 and 54.29, respectively [21]. However, the RR genotype was found to show disease occurrence in both field assessment and greenhouse inoculation, with infection rate of 3.4% and disease index of 1.57 [16]. The differences in the infection type and severity of the same *Z. japonica* inoculated were likely due to factors such as the incubation temperature and humidity, the concentration of the inoculant, the location of the inoculation, the virulence of the pathogen, and the duration of the pathogen in storage.

Morphological observations after inoculation with rust pathogen. The observation of pathogen infection process is very important to understand the interaction mechanism between host plant and pathogen [32]. To date, there have been no reports on the process of pathogen infection of zoysia rusts, and the ultrastructural changes of zoysia rusts in the period of their resistance to the disease are not clear. In this study, no obvious differences in cell organization or the degree of stomatal opening were observed at 0 to 1 dpi in both RR and RS *Z. japonica* genotypes. However, the infection of rust pathogen spores was more rapid in the RS, with longer the germ tube of the spore even at the early of infection. With the extended periods of infection, no degradation of leaf epidermal cells was observed in the RR plants, and numerous conidiophores had spread across the leaf surface (Figures 3 and 4). The results indicated that the growth of rust mycelium was restricted in the RR genotype, contributing to the slower growth rate of the rust pathogen in this type of *Z. japonica*. Our results supported the finding that the development of mycelium and the rate of spreading, as well as the timing of spore production, were delayed in the disease-resistant maize [33].

There were some discrepancies with the results between SEM and visual observations in this experiment. We hypothesized that SEM provided more accurate results and could further reduce the time frame of disease occurrence. When symptoms were visually observed, spores may have already undergone many generations of reproduction in the leaves. This could miss counting some stages of disease occurrence after inoculation. Therefore, we could observe the infection process of the rust pathogen and the different stages during this process using SEM. However, we encountered some problems during observation. Some stages could not be found such as the infection of spore germ tube in stomata. At the peak of asexual reproduction, the observed number of spores did not correspond well to the timing. These problems could be due to the sampling time and the techniques used for observation. Future studies should be cautious when observing the infection process.

Antioxidant enzymes and the corresponding gene expression. The SOD, CAT, POD, and APX are important enzymes that protect the cells from reactive oxygen species (ROS) damage and play important roles in plant defense mechanism under environmental stress [34]. In the ROS scavenging mechanism, SOD is the first enzyme that is activated

to be involved in the responses by removing superoxide radical [35]. In the present study, SOD activity in the RR genotype exhibited a higher increase than in the RS genotype at 5 to 15 dpi, suggesting a stronger capacity of scavenging superoxide and protecting cell in the RR plants. In antioxidant pathway, CAT and APX are responsible for decomposing hydrogen peroxide. The RR plants maintained higher CAT activity than RS plants by the end of infection, but not APX activity (Figure 6). In melon varieties after inoculating with powdery mildew, it took longer for CAT activity to increase in the susceptible variety, compared to the resistant one [36]. The results indicated that CAT activity could play a role in disease defense system in some plant species including *Z. japonica*. POD can catalyze the oxidation of phenols, promote lignin biosynthesis in plants, and increase plant disease resistance [37]. A previous study found that POD activity varied significantly in pepper varieties with different disease resistance [38]. In this study, from 5 to 10 dpi, POD activity increased in the RR genotype, with a higher peak than that in the RR genotype. Collectively, antioxidant enzymes play a role in increasing plant rust disease resistance, but some variations in activity occur due to duration of inoculation, severity of disease occurrence, and plant species.

The observed changes in CAT, POD and APX enzyme activity generally corresponded to their gene expression levels. The expression levels of *ChlCu/ZnSOD*, *MnSOD*, and *APX* were higher in inoculated RR plants and lower in RS plants than the non-inoculated plants, while the expression levels of *CAT* and *POD* were lower in the inoculated than the non-inoculated plants (Figure 7). Cu/ZnSOD enzyme exists in chloroplast and cytoplasmic matrix of higher plants [39,40]. MnSod enzyme, mainly located in mitochondria and peroxisome, which is responsible for removing ROS from mitochondrial matrix and protecting some biomolecules in mitochondria from oxidative modification [41]. In this experiment, the gene expression of *Cu/ZnSOD* and *MnSOD* in each treatment group was different, and indicated that the change of transcription level was related to the spatial location of the isoenzyme encoded by the gene. Also, the expression of *Cu/ZnSOD* and *MnSOD* genes in RR after inoculation were down-regulated, while the SOD activity was increased. This suggests that SOD activity might be regulated by other genes besides *Cu/ZnSOD* and *MnSOD*, but may be also influenced by post-transcriptional processing and other environmental factors during translation [42]. Further investigations are needed to study the comprehensive patterns of gene regulation of antioxidant enzymes for rust resistance.

## 5. Conclusions

In summary, morphological observation of spores and 18S rDNA molecular identification of the pathogen of zoysiagrass rust identified *P.zoysiae* Diet as the pathogen of rust in *Z. japonica* in Jiaozhou, Shandong Province. We further identified the RR and RS genotype based on their capacity of disease resistance. The expansion of the rust fungus was restricted by necrosis of the infected host cells and their adjacent cells, thereby causing a difference in infection between the RR and RS materials. Disease resistance was positively correlated with changes in antioxidant enzyme activities. The alterations of activities of CAT, POD, and APX were generally consistent with their gene expressions. The research results provides a basis for the selection of disease-resistant materials and the control of zoysiagrass rust.

**Supplementary Materials:** The following are available online at https://www.mdpi.com/article/10.3390/agriculture11121200/s1. Figure S1: The classification standard of rust disease severity in *Zoysia japonica* (Gu 2013), Figure S2: The classification standard of the infection type of rust diseases in *Zoysia japonica* (Gu 2013), Figure S3: PCR amplification of 18S ribosomal DNA (rDNA), Figure S4: Diagram of comparative evolution between 1811 and other rust fungi of Pucciniales based on the 18S ribosomal DNA (rDNA) sequence, Figure S5: Alignment of 18S ribosomal DNA (rDNA) gene sequence between 1811 and *Macruropyxis fraxini* (99%). The red-highlighted letters represent differences between *Macruropyxis fraxini* and 1811, Figure S6: 18S ribosomal DNA (rDNA) gene sequence alignment between 1811 and *Puccinia physalidis* (98%). The red-highlighted letters represent

differences between *Puccinia physalidis* and 1811, Table S1: Primer sequences for polymerase chain reaction, Table S2: Observation indexes of rust diseases in zoysiagrass (Gu 2013), Table S3: The infection type classification standard of rust diseases in zoysiagrass, Table S4: Primer sequences for qRT-PCR.

**Author Contributions:** Formal analysis, D.Z., L.Z., J.Y. and X.Q.; Project administration, M.L.; Supervision, M.L.; Writing—original draft, D.Z. and H.Z.; Writing—review & editing, M.L. All authors have read and agreed to the published version of the manuscript.

**Funding:** This research was partially financially supported by a Grant from the National Natural Science Foundation of China (No. 31101757) and China Agriculture Research System (CARS-034).

**Institutional Review Board Statement:** Not applicable.

**Informed Consent Statement:** Not applicable.

**Data Availability Statement:** The datasets used and/or analyzed during the current study are available from the corresponding author on reasonable request.

**Conflicts of Interest:** The authors declared that they have no conflict of interest to this work.

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
