# Peer review of "Isolation and Identification Rust Pathogens and the Study of Antioxidant Enzyme Activity and Gene Expression under Rust Infection in Zoysia japonica"

_agriculture, doi:10.3390/agriculture11121200_

Round 1

Reviewer 1 Report

The work submitted for review is interesting. I think it can be published with a little modification.  First of all, the work has many punctuation faults, lowercase/uppercase spelling of Latin names should be corrected. Moreover, it is worth adding a summary as a separate chapter.

Reviewer 2 Report

The manuscript describes an interesting study with both theoretically and practically important results. It is relatively well written, and only minor changes are necessary.

Title

Needs to be corrected to better reflect contents of the study, possibly including enzyme activity and expression as related to mechanism of tolerance.

Introduction

References are missing in the second paragraph (lines 35–37).

Make new paragraph on line 53 as new thought starts here.

In line 56, it is suggested to speak specifically on ascorbate peroxidase as antioxidative enzyme instead of peroxidase. However, peroxidase and polyphenol oxidase (which needs to be defined in full) are general markers of oxidative activity.

In line 50, use the full name of the genus, Zoysia.

Reference is missing in line 64.

In line 71 give full scientific name of the pathogen.

The aim needs to be formulated before short description how this aim was fulfilled.

Materials and methods

What was source of seeds? Cultivation system and conditions need to be described in detail (substrate, photoperiod, PAR, temperature, relative humidity).

What was age of plants at inoculation? What were biological replicates and in which number?

More details are necessary on obtaining homogenous material for enzyme and expression analysis (if only 0.1 g per analysis was used), extraction and measurement procedures. Number of replicates used?

In 2.2. indicate source names of both lines, as these are further used in Table 1.

It is difficult to understand why rust resistant material was indicated as RS and rust susceptible as SS. I suggest using more logical abbreviations, RR and RS, to avoid confusion.

Results

There is a mistake in column "Material" of Table 1, as both materials are indicated as SS.

In 3.5. first present actual results of enzyme activity (Figure 7) followed by their statistical analysis (Table 2). Include both residuals and F in Table 2. Use colours for Figure 7 and larger symbols. Indicate number of biological replicates and meaning of bars (SE or SD) in the legend for Figure 7 and Figure 8.

Not all photographs in Figures 5 and 6 seem to be of appropriate quality. Consider replacing with the better ones and make a compound picture with two columns (resistant vs susceptible) for comparison of identical times in dpi.

Discussion

The discussion is in line with the results obtained.

Only, I would like to suggest more detailed discussion on differences in activity and expression profile of different enzymes as related to their functions. Also, in respect to multigene existence and various isoforms.

Consider giving references to respective tables and figures when analysing results of the present study.
